# Revisiting evolutionary trajectories and the organization of the *Pleolipoviridae* family

**Tomas Alarcón-Schumacher** ⓘ *, **Dominik Lücking, Susanne Erdmann** ⓘ *

Max-Planck-Institute for Marine Microbiology, Bremen, Germany

* t.alarcon.sch@gmail.com (TA-S); serdmann@mpi-bremen.de (SE)

## Abstract

Archaeal pleomorphic viruses belonging to the *Pleolipoviridae* family represent an enigmatic group as they exhibit unique genomic features and are thought to have evolved through recombination with different archaeal plasmids. However, most of our understanding of the diversity and evolutionary trajectories of this clade comes from a handful of isolated representatives. Here we present 164 new genomes of pleolipoviruses obtained from metagenomic data of Australian hypersaline lakes and publicly available metagenomic data. We perform a comprehensive analysis on the diversity and evolutionary relationships of the newly discovered viruses and previously described pleolipoviruses. We propose to classify the viruses into five genera within the *Pleolipoviridae* family, with one new genus represented only by virus genomes retrieved in this study. Our data support the current hypothesis that pleolipoviruses reshaped their genomes through recombining with multiple different groups of plasmids, which is reflected in the diversity of their predicted replication strategies. We show that the proposed genus *Epsilonpleolipovirus* has evolutionary ties to pRN1-like plasmids from *Sulfolobus*, suggesting that this group could be infecting other archaeal phyla. Interestingly, we observed that the genome size of pleolipoviruses is correlated to the presence or absence of an integrase. Analyses of the host range revealed that all but one virus exhibit an extremely narrow range, and we show that the predicted tertiary structure of the spike protein is strongly associated with the host family, suggesting a specific adaptation to the host S-layer glycoprotein organization.

## Author summary

Pleomorphic viruses of the *Pleolipoviridae* family are abundant in hypersaline environments and are currently known to infect diverse members of halophilic archaea. Pleolipoviruses are able to establish chronic infections during which both, viruses and cells, are able to replicate and coexist, a rather rare trait for prokaryotic viruses. Our understanding of their diversity and evolution comes from a handful of isolates and remains scarce. In this study, we screened hypersaline lakes in Australia and public available databases and uncover novel pleolipoviruses, expanding the current known representatives by 10 fold, including new lineages composed of only uncultivated representatives. Furthermore, we propose a revised taxonomic organization for the *Pleolipoviridae* family, and shed light on

**Data Availability Statement:** Raw reads from metagenomic data generated in this study were submitted to ENA-EMBL under project number

PRJEB61734. All curated and annotated pleolipovirus genomes, as well as protein models for the spike protein can be found in https://doi.org/10.5281/zenodo.8248829.

**Funding:** The Max Planck Research Group 'Archaeal Virology' headed by S.E. is funded by the Max Planck Society, Munich. The funders had no role in study design, data collection and analysis, decision to publish, or preparation of the manuscript.

**Competing interests:** The authors have declared that no competing interests exist.

the evolutionary relationships and trajectories between the different clades. Our results show that pleolipoviruses have frequently recombined with other mobile genetic elements and that their replication strategies and genomes can be very flexible. Altogether, this study sets the basis for the discovery and characterization of new pleolipoviruses and their strategies to establish chronic infections in archaea.

## Introduction

Archaeal viruses represent one of the most fascinating part of the virosphere. Despite the low number of representatives, when compared to bacterial and eukaryotic viruses, they represent an unparalleled genomic structure, gene content and morphological diversity [1]. While some icosahedral viruses have been shown to be evolutionary related to bacterial and eukaryotic viruses [2,3], the vast majority of archaeal virus groups are unique to archaea and no relatives have been identified in the other domains of life [4]. Moreover, archaea-specific groups of viruses display distinct evolutionary origins, highlighting that viruses likely evolved on numerous multiple independent occasions [5–7]. Virus genomes typically exhibit structural and replication modules, which frequently recombine with each other and with a variety of mobile genetic elements (MGEs) such as transposons and plasmids [8,9].

One of the most intriguing groups of viruses demonstrating evolutionary mixing with different viruses and MGEs is the *Pleolipoviridae* family [3]. Viruses belonging to the *Pleolipoviridae* family, known as pleolipoviruses, are unique in many aspects, with members of this family shown to have different genome types, going from single or double stranded DNA genomes, to even hybrid double stranded genomes with single-stranded interruptions [10,11]. Unlike most prokaryotic viruses, which have a protein shell (capsid), the pleolipovirus virions have a host-derived lipid membrane enclosing their genome. Biochemical analyses have shown the presence of at least two major structural proteins in virions: a spike protein anchored to the membrane and a membrane associated protein facing the particle interior, commonly known as VP4 and VP3-like proteins, due to their position in the genomes of the first isolated representative Halorubrum pleomorphic virus 1 (HRPV-1) [10,12,13]. Virions are pleomorphic particles that range from 50 to 100 nm, which bud off from host cells without causing cell lysis. During this, so called persistent or productive chronic infections, virions are released in large amounts and have the potential to strongly impact host metabolism [10,14]. The majority of currently isolated pleolipoviruses were shown to be host specific, with the sole exception of recently characterized Haloferax pleomorphic virus 1 (HFPV-1), which displays an unusual broad host range, being able to infect members of both the *Haloferacaceae* and *Haloarculaceae* families [14].

Currently, pleolipoviruses are classified into three genera: the *Alpha*, *Beta* and *Gammapleolipovirus*. While the three genera share a stable core of genes consisting of two structural proteins, (i.e. VP3 and VP4 spike protein), plus two hypothetical proteins flanked by the spike protein and a conserved ATPase (ORFs 6 and 7 in HRPV-1), they exhibit a remarkably modular mosaicism in their genomes. This is particularly evident with their replication mechanisms. Viruses from the *Alphapleolipovirus* genus encode two non-orthologous families of rolling circle replication initiation endonuclease (RCRE), which were likely acquired by an ancestral pleolipovirus-like entity in independent events from two different groups of plasmids (pGRB1-like pTP2-like plasmids) [15]. Meanwhile, the *Betapleolipovirus* encode for a putative replication protein (Rep protein), that shows no homology to any other known replication protein, and its evolutionary origin remains a mystery [4]. On the other hand, the

*Gammapleolipovirus* include two viruses infecting *Haloarcula hispanica*, His2 and Hardyhisp2, with both encoding for a protein-primed DNA polymerase related to the spindle-shaped virus His1, which also infects the same host [16,17].

Recent metagenomic studies and the isolation and characterization of divergent members of the *Pleolipoviridae* family, as well as the identification of several pleolipovirus-like uncultivated viral genomes integrated in haloarchaeal genomes, have hinted that the diversity of this clade might remain largely unexplored [14,18–20]. Particularly, HFPV-1 exhibits low levels of sequence similarity with other isolated pleolipoviruses and encodes no homolog for any of the replication proteins, raising further questions on the diversity and organization of this clade. Thus, we aimed to expand our knowledge on the diversity of pleolipoviruses and study in depth the phylogenetic relationships between pleolipoviruses and other viruses and MGEs.

## Results and discussion

### Recruitment of 164 new pleolipovirus-like elements

To expand our understanding of the evolutionary relationships of the *Pleolipovirirdae* family, and improve our ability to detect potential novel members, we assembled a dataset consisting of the 17 previously isolated pleolipoviruses and searched for uncultivated viral genomes (UViGs) in the IMG/VR database. Selected genomes were then screened to identify the conserved cluster of four genes shared by all pleolipoviruses (ORFs number 4, 6, 7 and 8 in Halorubrum pleomorphic virus 1 (HRPV-1) [14] using homology annotation and HMM profile generation, profile validation, iterative search, and synteny verification. The gene arrangement of resulting contigs were further assessed for those containing the four conserved genes no more than 4.5kb apart and in the same order as observed in known isolates. This yielded 123 high-confidence UVIGs that were selected for further genome comparisons and phylogenetic analyses.

Given that the divergent pleolipovirus HFPV-1 was isolated from enrichment cultures of salt crust from Lake Tyrrell [14], we searched metagenomic data of Lake Tyrrell as well as 10 additional Australian salt lakes generated in this study (Fig 1), that were sampled during Australian summer 2018/2019, to further search for novel pleolipovirus genomes. To identify pleolipovirus-like sequences in the metagenomes, we screened individually binned and non-binned assembled data for the presence of the hallmark genes described above. This resulted in 261 contigs containing at least one of the conserved genes (minimum length 2kb) (see S1 Text for further details). In order to further deepen our understanding the taxonomic composition and the phylogenetic relationships of the mined pleolipoviruses, selected contigs were combined with the dataset of isolated and UVIG pleolipovirus-like genomes. Subsequently, genomes were clustered at 95% similarity (equivalent to the species level) and only those containing the four conserved gene cluster and a minimum length of 4.5kb, were considered for phylogenomic analyses [21]. This resulted in 184 non-redundant pleolipovirus genomes with an overall average completion of ~81% (17 isolates genomes, 126 UVIGs from the IMG/VR database and publicly available datasets and 41 genomes from Australian salt lakes metagenomic data generated in this study). Out of the 184, 100 genomes (~ 54%) were considered complete (see methods), with an average length of 13.5kb (S1 Table), which is considerably larger than the average genome length of current isolates (10.4kb). However, completeness and quality values estimated with checkV are likely underestimated, as several complete genomes containing terminal repeat or with genome length around the average were labeled incomplete and only medium quality given their extremely divergent gene content. Interestingly, all genomes, including the ones from this study and UVIGs from the IMGVR database, derive from hypersaline environments, indicating that pleolipovirus-like elements are

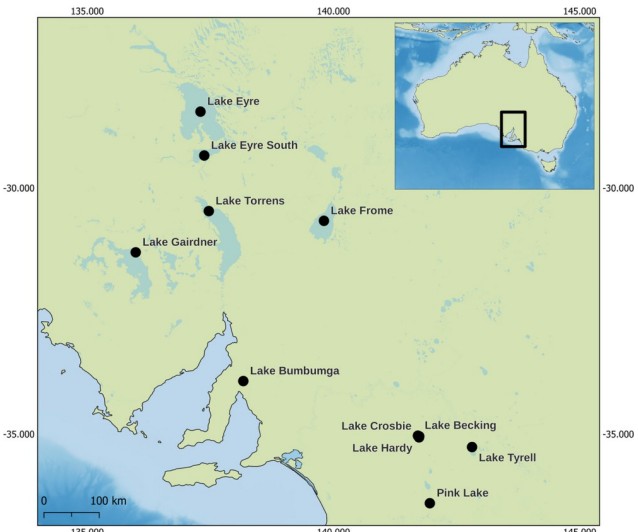

**Fig 1. Sampling sites.** Hypersaline lakes from south east Australia (South Australia and Victoria states) sampled for this study. Samples were collected during the austral summer of 2018/2019. Map data was obtained from public available databases Natural Earth, OpenStreetMap (https://www.openstreetmap.org/copyright) and the General Bathymetric Chart of the Oceans- GEBCO Compilation Group (2023) GEBCO 2023 Grid (doi:10.5285/f98b053b-0cbc-6c23-e053-6c86abc0af7b).

restricted to high salt environments. However, our inability to detect them in other environments could be caused by the low sequence similarity values (S1 Fig) and the dramatic variation in gene content among pleolipoviruses. Additionally, recent work has suggested that some pleolipo-like viruses could be part of the human gut virome of methanogenic archaea [22], further expanding their habitat distribution and suggesting that the relevance of this virus group might be underestimated.

## Genome wide phylogenomic organization of pleolipoviruses reveals unexpected high diversity

Given the extremely low similarity at nucleotide level between the isolated pleolipoviruses (S1 Fig), phylogenetic relationships between members of this clade have been previously studied using genome-wide based approaches [7,23]. Thus, we used the predicted proteins sequences and constructed a genome-based phylogeny with VICTOR [24] (Fig 2A). The majority of the retrieved viruses are related to the *Alpha* and *Betapleolipovirus* genera, with the *Alphapleolipovirus* forming a monophyletic group (although no significant support was obtained for the clade; score = 41). Most genomes within this group encoding for a homolog of a rolling circle replication endonuclease (RCRE), which is consistent with the current classification as the RCRE is one of the hallmark genes of the *Alphapleolipovirus* genus (ORF1 in HRPV-1) [23] (Fig 2A). Conversely, despite the apparent coherence with the topology previously proposed for the *Betapleolipovirus* genus, the vast majority of clades observed in the VICTOR tree present extremely low support values (< 95) indicating that the evolutionary relationships within this clade remain poorly resolved (Fig 2A). Despite sharing several characteristic proteins that serve as discrimination criteria for this genus (i.e. Halorubrum pleomorphic virus 3 ORFs 6 and 9), the *Betapleolipovirus* seem to form a polyphyletic group. They cluster into two major clades separated by several branches which consistently do not share these conserved features

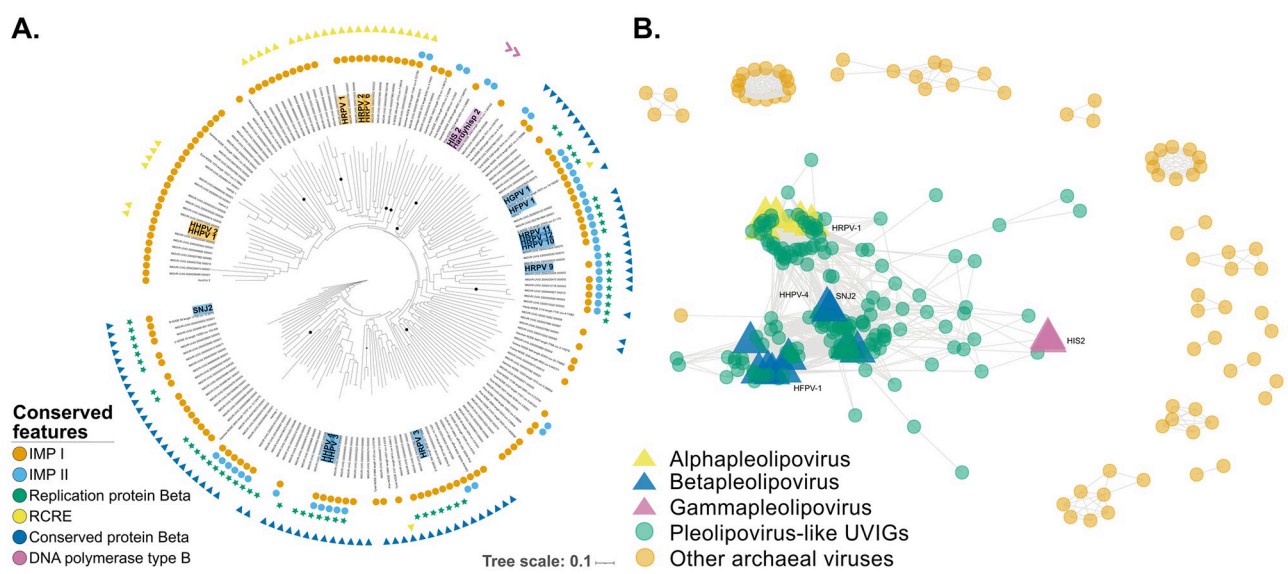

**Fig 2. Genome-wide phylogenetic analysis. A.** Phylogeny of pleolipoviruses generated with the Genome BLAST Distance Phylogeny (GBDP) algorithm implemented in VICTOR using amino acid sequences. The branches are scaled and represent the GBDP interproteomic distance inferred using formula D6. Supported branches (>95) are demarcated with black circles. Colored labels represent isolated representatives from each genus: *Alphapleolipovirus* (yellow), *Betapleolipovirus* (blue), *Gammapleolipovirus* (magenta). **B.** Gene-shared network of pleolipoviruses generated with vConTACT2. Nodes (circles) represent genomes and edges (lines) indicate shared protein content. Highlighted nodes are isolated representatives from the different genera: *Alphapleolipoviruses* (yellow), *Betapleolipoviruses* (blue) and *Gammapleolipoviruses* (magenta). Minimum shared protein clusters were set to two (—mod-shared-min 2).

(ORF6, ORF9), suggesting that they could be part of previously undescribed genera. Similarly, several genomes sharing no characteristic feature with neither *Alpha* nor *Betapleolipovirus* appear to be branching close to the *Gammapleolipovirus*. However, none of them encodes a type B DNA polymerase, while sharing only the core genes and sparsely hypothetical proteins with the other members of the *Gammapleolipovirus* genus.

Additionally, gene-sharing network analyses implemented in vConTACT2, which generates cluster that are equivalent to genera level classification [25,26], also revealed that not all genomes fit in the current trichotomy, with the analysis revealing 14 cluster with at least two members (S2 and S3 Figs). However, almost one third of the genomes (62 genomes) failed to fit into any of these genera-like groups, because these genomes shared a large proportion of their proteins with multiple clusters or were classified as singletons or low confidence pairs. To account for the extreme divergence at the sequence level and the smaller genome size of pleolipoviruses (relative to most viral taxa used to calibrate the genus-level classification), the analysis was repeated with a threshold of two-shared gene clusters (default 3 clusters in S3 Fig). This did not changed the overall topology of the network, however, it allowed us to establish connections between the main clusters and the most divergent genomes previously regarded as singletons or unconnected pairs (Fig 2B). Similar to the result observed with VICTOR, the network analysis confirms the existence of three main clusters that do not correspond with the three currently described genera (*Alpha*, *Beta* and *Gammapleolipovirus*). The *Alphapleolipovirus* genus cluster as a single group, however, the *Betapleolipovirus* form at least two different cluster. One cluster contains the small-genome isolates HFPV-1, HGPV-1 and HRPV-10 to 12 (approx. 8-9kb) and the other contains amongst others the larger-genome isolates HHPV-3 and HRPV-3 and the integrase encoding isolates HHPV-4 and SNJ2 (Fig 2B).

Pleolipoviruses also display little to no connections with other archaeal viruses, with the only exception *Yingchengvirus* SNJ1 virus, a member of the *Simuloviridae* family, whose members infect halophilic archaea, and that shares the host with pleolipovirus SNJ2 [27,28]. The absence of shared features with other viruses has also been observed for some viruses infecting hyperthermophilic archaea, and it has been suggested that this is likely a product of distinct evolutionary origins [6]. Altogether, the presented evidence suggests that the vast majority of proteins encoded by members of the *Pleolipoviridae* family are not shared with other archaeal viruses, and must have been obtained rather early in their evolutionary history or exchanged through recombination with other MGEs that infect the same host.

## Network analysis reveals a correlation between genome length and integration

Furthermore, the integrase encoding HHPV-4 and SNJ2 pleolipoviruses are among the nodes with the highest degree of connectivity (node degree of 70 and 69 respectively, Fig 2B), and interestingly, overall comparisons showed that genomes encoding integrases, have in general significantly higher degrees of connectivity than those which do not (Wilcoxon test, p-value = 0.00158). Further analysis revealed that, irrespective of their clustering patterns, pleolipoviruses carrying integrases also have significantly larger genomes (Wilcoxon test, p-value = $2.2 \times 10^{-16}$). On the other hand, no significant correlation was observed between node degree and genome length (correlation coefficient = 0.1 and 0.12 for Pearson and Spearman correlation tests respectively). While pleolipoviruses with the ability to integrate into their host genomes seem to harbor a more extensive gene repertoire, the connectivity between nodes is likely a result of their phylogenetic relationships and not random evolutionary convergence as a byproduct of larger genome sizes. Even though we cannot rule out that some viruses without integrases can integrate into their host genomes, as it has been shown for ssDNA viruses that employ alternative mechanisms for integration [29], the data suggests that pleolipoviruses rely on a self-encoded integrase to expand their genomic repertoire and to reliably fix the acquired genes in the population.

## Single gene phylogeny allows to resolve the evolutionary history of pleolipoviruses

To better resolve the evolutionary relationships between pleolipoviruses, we considered a concatenated phylogenetic approach using the four described core genes. They are conserved across all genera, indicating that they were already present at the origin of this virus group. To ensure that they display a congruent evolutionary history, we first performed single-gene phylogenetic reconstruction for each marker and compared the resulting topologies.

The phylogenetic reconstructions of ORF4 and ORF6 (in HRPV-1 genome), the structural spike protein and a hypothetical protein, displayed markedly different topologies (Fig 3), with little similarity to both the current taxonomic classification and the clades described here (Fig 2). The three genera (*Alpha*, *Beta*, *Gamma*) exhibit a polyphyletic nature, indicating that selective pressures that do not necessarily reflect the phylogenetic relationships between the different clades might be driving the clustering pattern of each of these proteins (see more in '*Host-driven evolution of the structural proteins*').

In contrast, ORF7 and ORF8 (in HRPV-1 genome) displayed a congruent topology (Fig 3), which is consistent with the higher sequence similarity of these two genes across pleolipoviruses genomes. The overall topology showed that the *Alpha* and *Gammapleolipoviruses* form monophyletic clades with high support values (SH-aLRT > = 80% and UFboot > = 95%). Meanwhile, beta-like pleolipoviruses organize into two separated well supported monophyletic

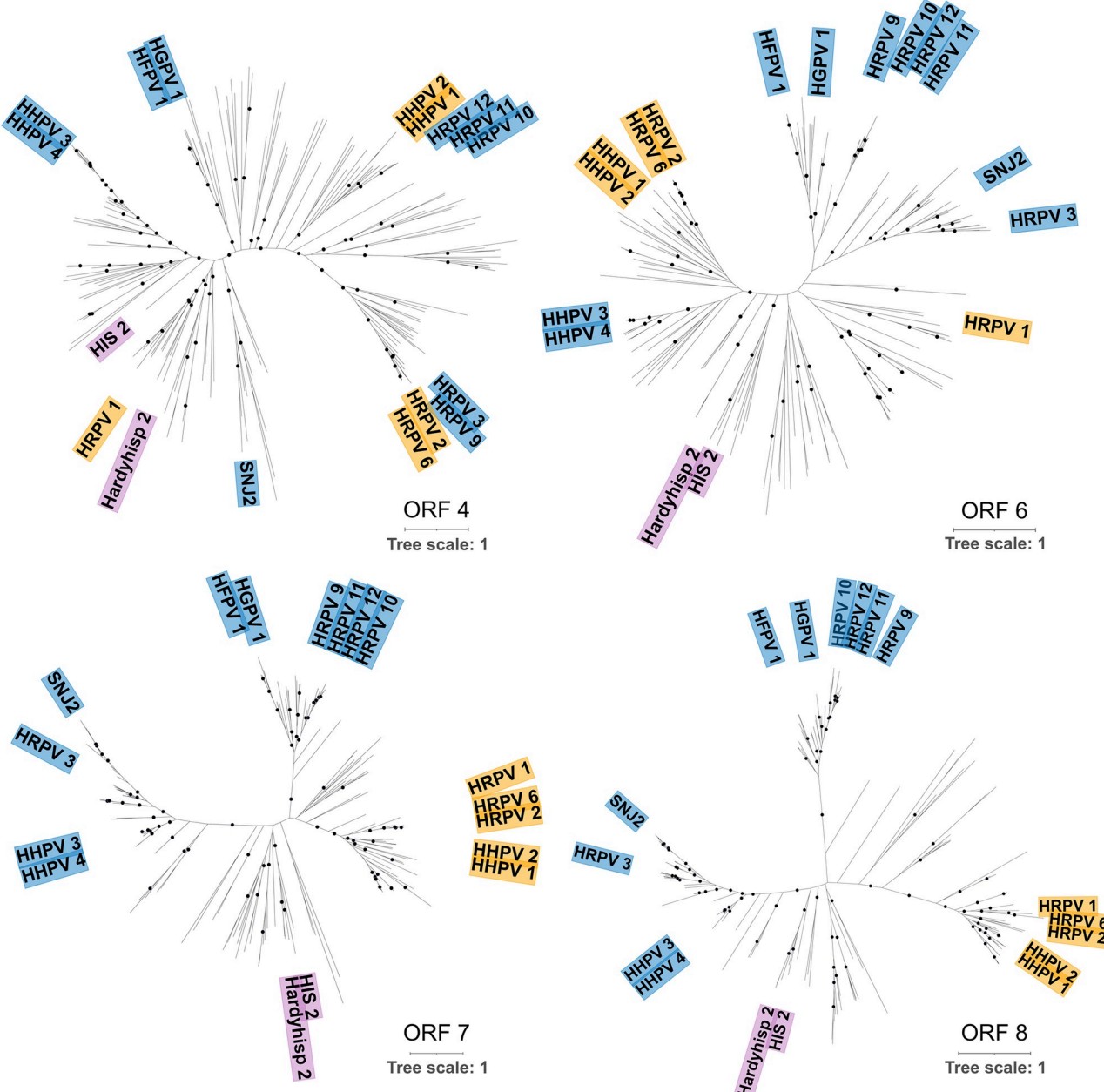

**Fig 3. Single gene phylogenies.** Phylogenetic tree reconstruction of conserved pleolipoviruses proteins. Sequences were obtained from the 184 pleolipovirus-like genomes dataset generated in this study. ORFs 4, 6, 7 and 8 correspond to the spike protein, two proteins of unknown function and an ATPase respectively, with ORFs numbers designated according to the annotation of Halorubrum pleomorphic virus 1 (HRPV-1). Tree was constructed with iqtree with 10.000 ultrafast bootstrap. Supported branches (SH-aLRT > = 80 and ultrafast bootstrap > = 95) are demarcated with black circles. Colored labels represent isolated representatives from each genus: *Alphapleolipovirus* (yellow), *Betapleolipovirus* (blue), *Gammapleolipovirus* (magenta). Scale bar represents the number of substitutions every 100 amino acids.

clades, consistent with the results obtained with VICTOR and the network analysis with vCon-TACT2, suggesting that they should be reclassified into separate lineages. Furthermore, using ORF7 and ORF8 as concatenated phylogenetic marker genes, resolves the relationships between previously poorly positioned branches (Fig 4). Additionally, the majority of genomes

# Proposed viral taxonomy

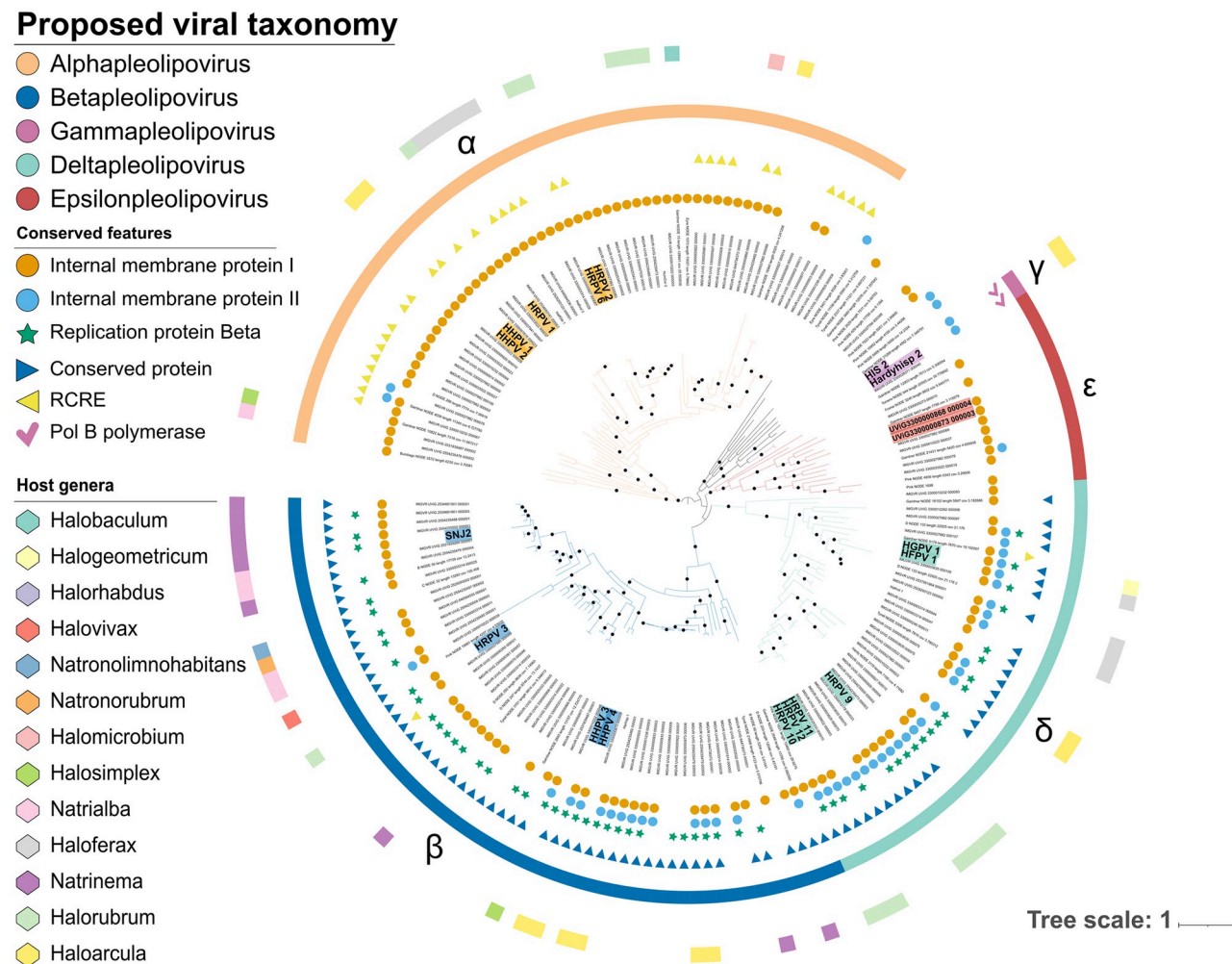

**Fig 4. Proposed taxonomy.** Phylogenetic inference from concatenation of ORF7 and ORF8-like genes using IQ-Tree. Supported branches (SH-aLRT > = 80 and ultrafast bootstrap > = 95) are demarcated with black circles. Greek characters represent different proposed genus within the *Pleolipoviridae* family: *Alphapleolipovirus* (α), *Betapleolipovirus* (β), *Gammapleolipovirus* (γ), *Deltapleolipovirus* (δ) and *Epsilonpleolipovirus* (ε). Scale bar represents the number of substitutions every 100 amino acids. From inside out the rings represent: presence of type I internal membrane protein (VP3-like), type II internal membrane protein, *Betapleolipovirus*-like Replication protein (Rep protein), Rolling circle replication endonuclease (RCRE), ORF8-like protein, Type B DNA polymerase, Proposed Taxonomy, Predicted Host genus.

that do not share the major hallmark genes with the established genera (i.e RCRE, polB polymerase, Rep protein, ORF8-like protein), form a well-supported separate lineage (aLRT > = 100% and UFboot > = 96.1%) (Fig 4). Meanwhile, a small number of genomes allocated close to the *Gammapleolipoviruses* remained persistent to classification, with many of them sharing little to none of the genes that currently serve as genera confining criteria [23]. Altogether, both genome-based, and single gene phylogeny, as well as the network analysis, indicate that they might be divergent members of the *Gammapleolipovirus* genus, or that they form separate clades.

## Proposed reclassification of the Pleolipoviridae with two new genera

Based on the presented evidence, we propose a new organization of the *Pleolipoviridae* family with the addition of two genera: *Deltapleolipovirus* and *Epsilonpleolipovirus* (Fig 4). This

reclassification includes the separation of the former *Betapleolipovirus* into two different lineages, the *Betapleolipoviruses* and the *Deltapleolipoviruses*, with the *Deltapleolipoviruses* including some members that were formerly classified as *Betapleolipoviruses* (i.e. HGPV-1 and HRPV-9 to 12). However, the *Deltapleolipoviruses* are a monophyletic clade that, despite sharing many features with the *Betapleolipoviruses*, diverged relatively early in the evolutionary history of pleolipoviruses (Fig 4 and S1 Text). Under the proposed taxonomy, the reported pleolipovirus genomes are distributed in the following genera: *Alphapleolipoviruses* (60 genomes, 40 complete), *Betapleolipoviruses* (59 genomes, 35 complete), *Gammapleolipoviruses* (two genomes, two complete), *Deltapleolipoviruses* (36 genomes, 19 complete), *Epsilonpleolipoviruses* (15 genomes, two complete), while 12 genomes remained unassigned.

The proposed new genus *Epsilonpleolipovirus* comprises only uncultivated members, with the complete genomes (UVIG) UViG3300000873 000003 and UViG3300000868 000004 chosen as the representatives for the clade (Fig 5). Consistent with the phylogenetic analysis, the most conserved genes are the genes equivalent to ORF7 and ORF8 in HRPV-1 with up to 60% similarity. Interestingly, the *Epsilonpleolipoviruses* possess a very flexible genome organization, as they display marked differences in gene content with each other and even some of their core genes share little to none protein sequence similarity (Spike protein and ORF6-like proteins) (Fig 5). The spike protein and ORF6 exhibit abnormally large sizes when compared to other pleolipoviruses, however, structural prediction of the spike protein, using the representative genome, reveals a similar v-shaped fold as reported for pleolipoviruses HRPV-2 and HRPV-6 [30] (Fig 6). A host could not be identified for any member of the *Epsilonpleolipoviruses*, indicating that they might be infecting different taxa than previously described pleolipoviruses, which could be the cause for the observed changes to the spike protein. Through additional structural prediction of neighbor proteins, we also identified a divergent variant of the VP3-like internal membrane protein (ORF6 in Pink_NODE_1698, ORF19 in UViG3300000873 000003, Fig 5), which despite the lack of sequence homology exhibits a highly similar predicted fold to the one predicted for the member of the sister clade *Deltapleolipovirus* HGPV-1 (S4 Fig).

The replication strategy of *Epsilonpleolipovirus* seems rather diverse (see also S1 Text). They do not share any of the hallmark replication genes with the other genera (RCRE, Rep protein nor type B polymerase). However, some (3 out of 14) encode for proteins with a DNA primase/polymerase domain (IPR015330, S2 Table), which has been described in archaeal plasmids, such as pRN1 from *Sulfolobus islandicus* [31], and highlights again the relevance of plasmid-virus recombination in the evolutionary trajectory of this clade. Another member (IMGVR_UViG_3300005273_000010) encodes for a non-related phage P4-like DNA primase (IPR006500, S2 Table). However, the remaining *Epsilonpleolipovirus* members, including three complete genomes, do not exhibit a homolog for any known replication protein, suggesting that they might replicate via a combination of host polymerases and virus-encoded replication factors. Most dsDNA archaeal viruses do not encode their own polymerase, instead, they encode proteins involved in replication initiation such as minichromosome maintenance (MCM) helicases, Orc1/Cdc6 proteins or a PCNA [32–34]. Additionally, 9 out of 14 *Epsilonpleolipovirus* genomes encode small proteins homologues to HNH-like nucleases (IPR003615, Fig 5). These nucleases are commonly found in archaeal and bacterial tailed viruses [35,36]. However, their role in the life cycle of *Epsilonpleolipoviruses* remains to be elucidated.

## Pleolipoviruses have a very narrow host range

The spike protein is crucial for host recognition, interacting with the host cell surface and triggering membrane fusion [37,38]. Interestingly, the phylogenetic reconstruction on the spike

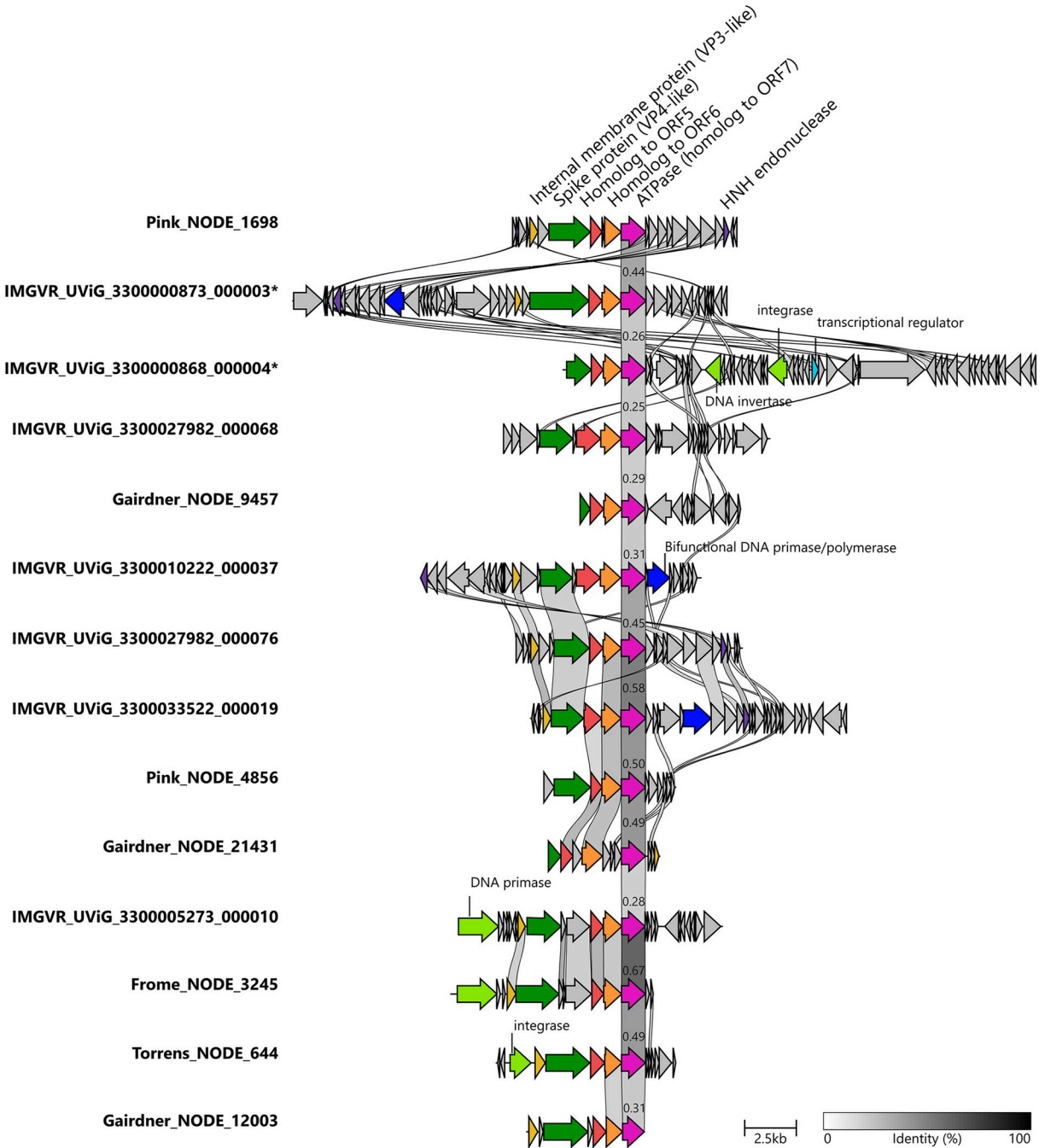

**Fig 5. The *Epsilonpleolipovirus* genus.** Genomic alignment of members of the proposed *Epsilonpleolipovirus* genus. Similarity values (blastp) are indicated by grayscale shading. Homologues of conserved genes are colored the same as follows: VP3-like protein (light brown), Spike protein (dark green), ORF6-like (red), ORF7-like (orange), ATPase (magenta), HNH-like endonuclease (purple), bifunctional DNA polymerase/primase (blue), transcriptional regulator (light blue) and other DNA-related enzymes, i.e. integrase, primase and invertase (light green). Complete genomes are highlighted with (*) symbol.

protein amino acid sequences (Fig 3, ORF4), displayed a clustering pattern more related to the taxonomic affiliation of the hosts rather than to the virus taxonomy. To explore the selective pressures driving the evolution of the spike protein, we performed an in silico host prediction for our entire dataset of pleolipoviruses using Iphop [39]. Iphop combines multiple approaches

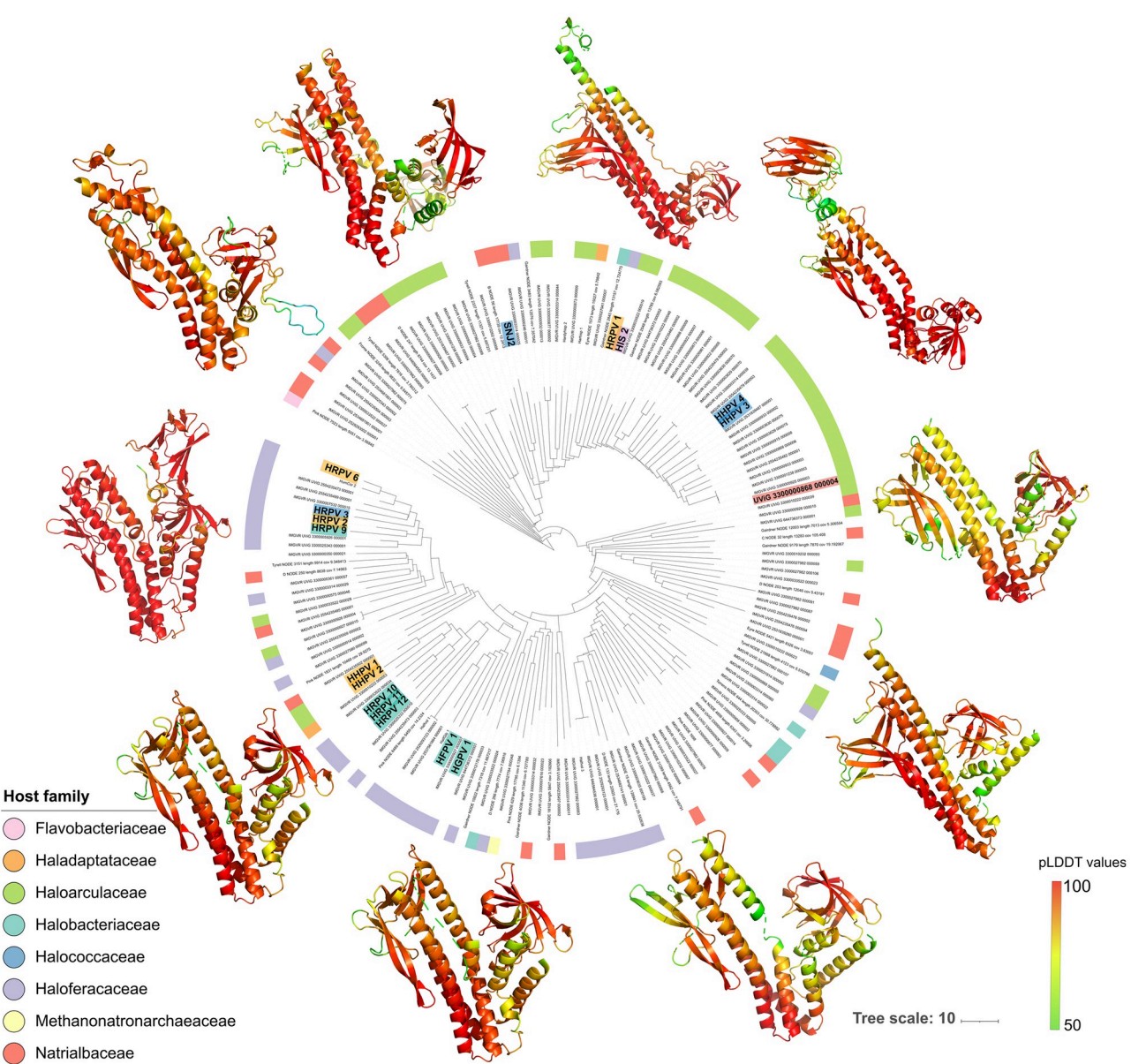

**Fig 6. Spike protein structural models.** Cladogram of structural models of pleolipovirus spike proteins. Pairwise structural similarities (Z) scores were calculated in DALI. Representative structures for major clades are shown using ribbon representation and colored according to the plDDT values. Colored labels indicate representatives from different genera: *Alphapleolipovirus* (yellow), *Betapleolipovirus* (blue), *Gammapleolipovirus* (magenta), *Deltapleolipovirus* (turquoise) and *Epsilonpleolipovirus* (red). Ring represent the taxonomic affiliation (at family level) of the predicted host for the respective virus.

to establish virus-host pairs, such as matching CRISPR spacers, protein content and k-mer frequency algorithms (see methods). The prediction yielded significant (Iphop score > 90) results for 125 of the 184 genomes (68%), out of which, 57 virus-host pairs (~30%) were deemed as certain, because they were identified as integrated proviruses or identified by CRISPR spacer matches (S1 Table). Notably, the vast majority of UVIGs were targeted by CRISPR spacers only from a single host organism at a time (S5 Fig), further confirming the proposed host specificity for members of this family [23]. These results correlate well with experimental data for

pleoliopviruses. Those shown to have a narrow host range, infecting only the strain from which they were isolated [10,40], are also targeted by spacers from single hosts. On the other hand, HFPV-1, which was shown experimentally to successfully infect distant-related host organisms, even crossing the family barrier [14], was targeted by CRISPR spacers from 13 different host species (S5 Fig).

For those viruses with a predicted host, almost 90% of the hosts belong to three families within the *Halobacteriales* order; the *Haloarculaceae*, *Haloferacaceae* and the *Natrialbaceae* family (39.2%, 31.2% and 20% respectively, S1 Table). The most represented genera are *Haloarcula* (28%), *Halorurbum* (16%), *Natrinema* (11.2%) and *Haloferax* (9.6%). All but one of the remaining UVIGs were associated with other genera within the *Halobacteriales* order. Surprisingly, one UVIG (Pink NODE 7523) was associated with the *Flavobacteriaceae* family (Iphop score 91.8), mainly supported by the protein content-based prediction implemented in RaFAH [41]. Interestingly, this UVIG (Pink NODE 7523) was retrieved from Pink Lake metagenomic data, and is related to the divergent *Gammapleolipovirus*-like genomes, for whom no host could be predicted (Figs 4 and S6). These UVIGs do not share any of the non-structural hallmark genes with any of the currently accepted nor proposed genera, with most of its proteins lacking a functional annotation. A pleolipovirus infecting a bacterial host has not yet been reported, and the morphologically similar bacteria-infecting phage mycoplasma virus L172 is not phylogenetically related to pleolipoviruses. [23,42,43]. It is therefore unlikely, but not impossible, that this pleolipovirus would infect a bacterial host. We suggest that this and other gamma-like genomes are likely divergent members within the *Pleolipoviridae* family that form their own individual lineages. In the particular case of the UVIG Pink NODE 7523, it could have evolved through recombining directly with some bacterial-related MGEs, or more likely, emerged as an indirect product of previous horizontal gene transfer (HGT) between bacteria and the archaeal host cell, as HGT has been well documented to occur often between these two domains [44–46].

## Host-driven evolution of the structural proteins

To further investigate the impact of the host specificity on the evolution of pleolipoviruses we performed in-silico structural prediction of all retrieved spike proteins using AlphaFold2 [47]. Despite the low levels of similarity at the amino acid sequence among UVIGs, structural comparison of the spike proteins revealed the conservation of the V-shaped fold previously reported for HRPV-2 and HRPV-6 spikes amongst all isolates and UVIGs (Fig 6) [30]. However, the predicted structures form two distinctive clades unrelated to the pleolipovirus taxonomic classification (Fig 6).

In the reported structures for HRPV-2 and HRPV-6 the overall V-fold structure is subdivided in two main domains (N and C terminal) of roughly equal size. The N-terminal domain is again subdivided in two subdomains: N1, composed of a bundle of 4 α-helices, and N2, composed of an array of beta sheets, with the latter being the proposed receptor binding domain [30]. While the N-terminal N1 subdomain was extremely conserved across all predicted structures, the N2 subdomain was less conserved (several cases of poorly resolved and low plDDT values). Conversely, confidently predicted structures displayed a wider arrange of conformations for N2 (Fig 6), which is likely a result of the monospecific nature of their host range acting as the major evolutionary force.

The structure of the C-terminal domain of the HRPV-2 and HRPV-6 spike proteins display three subdomains (C1-C3) preceding a ~20 amino acid transmembrane α-helix domain that anchors the spike protein to the membrane [30]. While the transmembrane domain was relatively conserved (S7 Fig), the C-terminal domain (C1-C3) appeared rather diverse, as observed

for the N2 receptor binding domain. Modeled spike proteins that clustered close with the structures from HRPV-2 and HRPV-6, displayed a similar three-subdomain organization. However, the rest of the spikes displayed mainly a 2-subdomain organization of the C-terminus, with one subdomain being similar to the C3 subdomain (2 to 3 α-helixes) and the second one to the C2 subdomain with beta sheets organized as small beta barrels [48]. The most divergent structures were found in the branch containing mainly pleolipoviruses predicted to infect members of the *Haloarculaceae* family (Fig 6). The modeled spike proteins in this branch exhibit the two short α-helixes C2 subdomain, and the small beta-barrel folded C1 subdomain in close proximity, while an unstructured chain of residues connects the two domains with the transmembrane domain (S8 Fig).

The overall clustering patterns of the structural dendogram seems to be mostly related to the host taxonomy, with the two main branches relating to the *Haloferacaceae* (classic V-shaped fold) and the *Haloarculaceae* family (shortened C-terminal fold) (Fig 6). Interestingly, the two most variable structural features, i.e. the receptor binding N2 subdomain and the C-terminal domain, are the ones that likely interact closer with the cell surface of the hosts. The archaeal cell is covered by the S-layer glycoprotein, which was shown to be recognized by the N2 subdomain of the spike protein of HRPV-6, and to trigger membrane fusion, the key process during the infection of a new host [38]. However, archaeal S-layers share little sequence similarity even between closely related species. This suggests that the high variability in the N2 subdomains is likely a product of host selective pressures and adaptations to allow initial interaction with a single S-layer protein from a specific host. Additionally, the S-layer likely also interacts with the C-terminal domain of the spike protein while it is anchored to the membrane. The S-layer of *Haloferax volcanii* and other members of the *Haloferacaceae* family was shown to be organized in a dome-shaped manner [49,50], while the triangular archaeon *Haloarcula japonica*, a member of the *Haloarculaceae* family, was proposed to have an arched shaped S-layer structure [50,51]. This suggests that the organization of the S-layer glycoprotein coat, whether it is a domed for *Haloferacacea*e or an arched organization for *Haloarculaceae*, might be driving the reported diversity of the C-terminal domain of the spike protein of pleolipoviruses. Altogether, we conclude that the spike protein of pleolipoviruses has evolved driven by the interaction with the S-layer glycoprotein of their specific hosts and it does not represent a reliable marker for the evolution of the *Pleolipoviridae* family. Nonetheless, the host-related clustering patterns observed with the structural comparison have the potential to become a valuable tool to help predict the hosts of yet uncultivated pleolipoviruses and other UVIGs.

## Conclusion

In this work, we presented 41 new archaeal pleomorphic virus genomes, which originate from 11 hypersaline lakes in southeast Australia, and mined publicly available datasets to perform a comprehensive study of the *Pleolipoviridae* family, expanding in more than one order of magnitude the current diversity of isolated pleolipoviruses [23]. Similarly to other archaeal viruses, pleolipoviruses display remarkably low sequence similarity, at both nucleotide and protein level, and like haloarchaeal tailed viruses, there seems to be little connection between their geographical origin and their taxonomic organization. Through multiple approaches, we showed that the current taxonomic classification results in several polyphyletic groups, and therefore, propose a new organization for the *Pleolipoviridae* family into five different genera: *Alpha*, *Beta*, *Gamma*, *Delta* and *Epsilonpleolipovirus*.

Moreover, we showed that the evolutionary history of this family has been repeatedly impacted by recombination events with different groups of MGEs. Particularly features shared with plasmids from diverse origins and environments suggests that these groups of viruses

could possibly be found in environments other than hypersaline environments, and that pleo-lipoviruses possibly infect organisms other than haloarchaea. Altogether, this work sheds light on the diversity and evolution of the *Pleolipoviridae* family, and lays the foundations for a better understanding of these chronic viral infections and their impact on host metabolism and ecology, which has been scarcely investigated.

## Materials and methods

### Sampling sites

Sediment salt crust were collected from a total of 11 hypersaline lakes in December 2018 and January 2019 (Fig 1) under the permission from the Department for Environment and Water, South Australia (Permission number: U26817-1) and the Department of Environment, Land, Water Planning, Victoria (Permission number:1008945). DNA was extracted from approximately 1g of sediment with FastDNA SPIN Kit for Soils. DNA sequencing libraries (FS DNA Library (NEBNext Ultra) and sequencing (Illumina HiSeq2500—Rapid Mode) was performed at the Max Planck-Genome-Centre Cologne (Germany), with run condition 2 x 250 bp (paired end reads). Reads were trimmed with Cutadapt [52], allowing a minimum quality of 30 and discarding short and unpaired reads (-q 30, -m 30). Quality-trimmed reads were assembled with metaSPAdes v3.13.1 [53] and protein prediction was performed using Prodigal [54] in metagenomic mode (-p meta). Contig coverage was calculated using BBmap v38.06 [55] with a minimum identity of 90% (minid = 0.9). Assembled contigs were then binned with MetaBAT 2 [56] allowing a minimum contig length of 1500 nucleotides (-m 1500), a maximum number of edges of 1000 to increase sensitivity (—maxEdges 1000) and a minimum score of 95 to increase specificity (—minS 95), with the—noAdd flag to diminish contamination issues. For exact reproducibility a seed was also indicated (—seed 1). Quality assessment of generated bins was performed with CheckM [57]. High and Medium-quality MAGs were determined according to the standards developed by the Genomic Standards Consortium [58] and selected for further analyses. Taxonomic classification of selected MAGs was performed with the GTDB-Tk toolkit v1.4.0 [59].

### Generation of Pleolipovirus database

First, we generated a trusted database for each of the proteins conserved in all the available sequences of 16 previously isolated pleolipoviruses (ORFS 4,6, 7 and 8 in HRPV-1) [14,18]. Then, we used the sequences corresponding to each gene to query the IMG/VR database version 5.1 [60,61] for related viruses using blastp implemented in Diamond [62]. Hits with an e-value $< 10^{-5}$ and a score $> 50$ were considered significant and added to the database of each core gene. This process was recursively iterated adding the new significant hits from each iteration to the respective database until no new hits were obtained. Then, Hidden Markov model (HMM) were generated for each gene, for which alignments were performed with MAFFT v7.407 [63] and the—localpair and—reorder flags. Subsequently, HMM profiles were generated with HMMER v. 3.2.1 [64]. HMM models for each gene were then used to further identify distant homologs for each gene using the same recursive approach described above. In order to reduce false positive results and to enhance specificity, HMM models were trained against a dataset containing a combination of protein sequences from identified pleolipoviruses (true positives), and an array of proteins of known function in the PFAM database v.35.0 [65] (true negatives). Hits with a minimum e-value of $10^{-5}$ and above the score threshold (score $> =$ to the lowest by a true positive and 10 units higher than the highest score observed for a true negative in the training dataset), were considered as true positives hits for each model.

## Identification of genomes in metagenomic data

Resultant models were then used to screen binned and non-binned contigs, generated in this study, to obtained potential virus genomes belonging to the *Pleolipoviridae* family. Contigs of at least 4.000 bp and with significant hits against at least four of the previously generated models of conserved genes among pleolipoviruses were considered for further analyses. Quality control to identify and exclude non-viral regions was performed with CheckV [66], database version 1.4 (Aug 27, 2022). Due to the limited representation of pleolipoviruses in CheckV database, and in order to properly asses the quality of potential pleolipoviruses-like genomes, the HMM models from conserved proteins previously generated in this work were added to the database and accounted as viral proteins. Additional identification of integrated proviral sequences and the presence of direct and inverted terminal repeats (DTRs and ITRs respectively) were performed with Virsorter2 and geNomad [67,68]. Genomes were then considered complete if they presented terminal repeats (TRs), were identified as complete integrated proviruses or if they displayed CheckV completeness value = 100. Quality-trimmed sequences were clustered at 95% nucleotide identity with the software NUCmer (NUCleotide MUMmer) version 3.1 [69], which corresponds to the species level [21,70].

## Genomic and phylogenetic analysis of pleolipoviruses genomes

For genomic characterization, viral contigs retrieved from metagenomic data were combined with genomes of previously isolated pleolipoviruses and those retrieved from the IMG/VR, and redundancy was revised with NUCmer as described above. Average nucleotide distance (ANI) was calculated using PyANI v.0.2.12 [71] and the VIRIDIC web server [72]. Functional annotation of predicted viral proteins was performed InterProScan v5 [73], with DRAM [74], the package HH-suite3 [75] against the PDB70 database (release pdb70 200108) and the BFD database [76]. Functional categories were assigned using eggnog-mapper [77,78].

Genome based phylogeny and classification was performed with VICTOR [24] using the predicted protein sequences and the D6 formula. Gene-shared network analyses were performed with vConTACT [25,26] using the Viral RefSeq-archaea database v.211, blastp as the relations mode (—rel-mode BLASTP) with a minimum e-value of $10^{-3}$, a significance threshold in the contig and protein cluster similarity network of 0.5 (—sig 0.5,—mod-sig 0.5) and with the minimum number of contigs a protein cluster must appear of 2 (—mod-shared-min 2). Network results were displayed using R packages ggplot2 and Network [79,80].

## Single gene phylogeny

Protein sequences from selected conserved genes were aligned using MAFFT v7.407 [63] with local pair strategy (—localpair). Alignments were trimmed with the Clipkit software using default parameters [81]. Trimmed alignments were used to infer phylogenetic relationships using IQ-TREE 2 [82] with 10.000 ultrafast bootstrap [83] and 10.000 replicates of SH-aLRT branch test and automatic model selection [84]. Results were then visualized using iTOL [85].

## Protein structural prediction

Protein structures were predicted using Alphalfold 2 [47] or retrieved from protein structure databases when available [86,87]. Protein structures were visualized with open source PyMOL Molecular Graphics System, Version 2.2.0 [88]. Resulting structural models were manually inspected and low-confidence (pLDDT < 50) N and C-terminal regions were removed. Subsequently, selected models were used for structural similarity comparison with DALI [89].

### Virus-host inference

Host prediction was performed with the integrated phage host predictor Iphop [39], which combines multiple approaches for virus-host prediction, i.e. blast to reference host genomes and CRISPR spacers database; k-mer composition algorithms implemented in tools WIsH, PHP and VHM—s2* [90–92]; and protein content-based prediction implemented in RaFAH [41]. In case of multiple hosts with significant scores ($> = 90$), hosts containing CRISPR matches against the virus were considered as true positives. If no CRISPR spacer was identified, the highest integrated Iphop score value was considered as the most likely host.

## Supporting information

**S1 Text. Supplementary Results and Discussion.**
(DOC)

**S1 Fig. Pleolipovirus genome similarity.** Average nucleotide identity (ANI) values were calculated with PyANI using MUMner as alignment method. Values are displayed as percentage identity (ANIm).
(TIF)

**S2 Fig. Viral clusters.** Pleolipovirus-like genomes were clustered based on their shared protein content with vConTACT2. Numbers on top of bars indicate the number of pleolipovirus genomes in the respective cluster. Colored labels represent isolated representatives from each genus: *Alphapleolipovirus* (yellow), *Betapleolipovirus* (blue), *Gammapleolipovirus* (magenta). Genomes classified as singletons, outliers and belonging to overlapping clusters were grouped under the category "Unclustered".
(TIF)

**S3 Fig. Pleolipovirus network analysis: Gene-shared network of pleolipoviruses generated with vConTACT2.** Nodes (circles) represent genomes and edges (lines) indicate shared protein content (minimum three protein clusters). Highlighted nodes are isolated representatives from the different genera: *Alphapleolipoviruses* (yellow), *Betapleolipoviruses* (blue) and *Gammapleolipoviruses* (magenta).
(TIF)

**S4 Fig. Internal membrane protein comparison.** Structure prediction of the internal membrane protein type I (ORF2 in Haloferax pleomorphic virus 1) generated with AlphaFold2. Representative structures the genera *Alphapleolipovirus* (HRPV-1), *Deltapleolipovirus* (HGPV-1) and *Epsilonpleolipovirus* (Pink_Node_1698) are shown using ribbon representation and colored according to the plDDT values.
(TIF)

**S5 Fig. CRISPR spacers targeting.** CRISPRs spacers were queried using blastn against the 184 pleolipovirus genomes from this study to assess the potential host range. **A.** Number of pleolipovirus genomes targeted by different hosts. **B**. Number of spacers from a specific host CRISPR array(s) for each one of the identified virus-host pairs. Numbers on top of the bars indicate the number of genomes and virus-host pairs respectively for A and B.
(TIF)

**S6 Fig. The gamma-like pleolipovirus.** Genomic alignment of genomes related to the *Gammapleolipovirus* genus. Similarity values (blastp) are indicated by grayscale shading. Homologues of conserved genes are colored the same as follows: Spike protein (dark green), ORF5-like (red), ORF6-like (orange), ATPase (magenta), type B DNA polymerase (blue),

and DNA methyl transferase (light green). Complete genomes are highlighted with (\*) symbol.
(TIF)

**S7 Fig. HMM logo spike protein.** Multiple sequence alignment of the spike protein. Sequences were obtained from the 184 pleolipovirus-like genomes dataset generated in this study and aligned using MAFFT v.7. Height of the letter depict the relative amino acid frequencies for each position of the alignment.
(TIF)

**S8 Fig. Spike protein structural comparison.** Structure prediction of the spike protein of the *Alphapleolipovirus* HRPV-6 and the *Gammapleolipovirus* His2 type I generated with Alpha-Fold2. Representative structures for major clades are shown using ribbon representation and colored using rainbow scheme from the N-terminus (blue) to the C-terminus (red). C1-C3 and N1-N2 indicate the protein subdomains.
(TIF)

**S9 Fig. Provirus genome.** Genomic representation of newly identified Halobaculum pleolipovirus-like provirus. Homologues of conserved genes are colored the same as follows: Rolling circle replication endonuclease (RCRE) (yellow), VP3-like protein (green), Spike protein (light blue), ORF5-like (purple), ORF6-like (magenta), ATPase (blue) and integrase (light brown). Insertion site is depicted by the tRNA corresponding to tryptophan.
(TIF)

**S10 Fig. Phylogeny conserved Rep Beta.** Phylogenetic tree reconstruction of the proposed replication-like proteins of *Betapleolipoviruses*. Sequences were obtained from the 184 pleolipovirus-like genomes dataset generated in this study. Tree was constructed with iqtree with 10.000 ultrafast bootstrap. Supported branches (SH-aLRT $>$ = 80 and ultrafast bootstrap $>$ = 95) are demarcated with black circles. Scale bar represents the number of substitutions every 100 amino acids.
(TIF)

**S11 Fig. Phylogeny conserved ORF8.** Phylogenetic tree reconstruction of ORF8-like proteins. Sequences were obtained from the 184 pleolipovirus-like genomes dataset generated in this study. ORF8 was designated according to the annotation of Haloferax pleomorphic virus 1 (HFPV-1). Tree was constructed with iqtree with 10.000 ultrafast bootstrap. Supported branches (SH-aLRT $>$ = 80 and ultrafast bootstrap $>$ = 95) are demarcated with black circles. Scale bar represents the number of substitutions every 100 amino acids.
(TIF)

**S1 Table. Pleolipovirus genomes compiled data.** Excel file of S1 Table summarizes the integrated data from the analysis performed in this study for each high-confidence pleolipovirus genome. Integrated completeness assessment is the result of the combination of the following criteria: Genomes were then considered complete if they presented terminal repeats (TRs), were identified as complete integrated proviruses, or if they displayed CheckV completeness value = 100. The Proposed Genus taxonomy column integrated the phylogenetic reconstruction of multiple marker genes (see main text results and discussion). Host prediction was performed with Iphop, and only results with a minimum confidence score of 90 are displayed. Topology values and main viral groups (single-stranded or double-stranded DNA) were obtained with Virsorter2. General information for a subset of only complete genomes are on the second tab under: arPVs_complete_genomes. Data on only the subset of genomes obtained from the metagenomic data produced in this study can be found in the tab

arPVs_Australian_lakes.
(XLSX)

**S2 Table. Functional prediction *Epsilonpleolipovirus*.** Excel file of S2 Table summarizes the functional prediction of the proteins from the proposed *Epsilonpleolipovirus* genus. Prediction was performed with interproscan v5 and functional categories were assigned using eggnog-mapper (minimum e-value $< 10^5$).
(XLSX)

**S3 Table. Pleolipovirus-like elements from Australian hypersaline lakes.** Excel file of S3 Table summarizes the completeness and quality assessment performed with CheckV for all pleolipovirus-like contigs detected in the metagenomic data generated in this study (minimum contig length 2kb). Average values of length, viral and total gene count, host genes, completeness and contamination are indicated at the bottom of each respective column. Database version 1.4 (Aug 27, 2022) was used.
(XLSX)

**S4 Table. Metagenomic assembled genomes from Australian hypersaline lakes.** Excel file of S4 Table summarizes the taxonomic affiliation and the quality assessment of metagenomic assembled genomes (MAGs) retrieved from the metagenomic data generated in this study. Completeness and contamination analysis were performed with CheckM. High and medium-quality MAGs were then selected for further taxonomic classification with the GTDB-Tk toolkit v1.4.0. High-quality MAGs are highlighted in green.
(XLSX)

## Acknowledgments

We thank Dr. Renate Dohmen from the Max Planck Computing & Data Facility, Garching, Germany for the infrastructure used for structural prediction with Alphafold2. We thank Daniela Thies and Ingrid Kunze (MPI for Marine Microbiology, Bremen, Germany) for assistance with some of the experiments. We thank Mart Krupovic and Hanna Oksanen for fruitful discussions. Finally, we want to thank the Max-Planck-Institute for Marine Microbiology and the Max-Planck-Society for continuous support.

## Author Contributions

**Conceptualization:** Tomas Alarcón-Schumacher, Susanne Erdmann.

**Data curation:** Tomas Alarcón-Schumacher.

**Formal analysis:** Tomas Alarcón-Schumacher.

**Funding acquisition:** Susanne Erdmann.

**Investigation:** Dominik Lücking.

**Methodology:** Tomas Alarcón-Schumacher.

**Supervision:** Susanne Erdmann.

**Validation:** Susanne Erdmann.

**Visualization:** Tomas Alarcón-Schumacher.

**Writing – original draft:** Tomas Alarcón-Schumacher.

**Writing – review & editing:** Dominik Lücking, Susanne Erdmann.

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
