## [Decision Letter · Decision Letter 0]

14 Jul 2023

Dear Dr Erdmann,

Thank you very much for submitting your Research Article entitled 'Revisiting evolutionary trajectories and the organization of the Pleolipoviridae family' to PLOS Genetics.

The manuscript was fully evaluated at the editorial level and by independent peer reviewers. The reviewers appreciated the attention to an important problem, but raised some substantial concerns about the current manuscript. Based on the reviews, we will not be able to accept this version of the manuscript, but we would be willing to review a much-revised version. We cannot, of course, promise publication at that time. 

Should you decide to revise the manuscript for further consideration here, your revisions should address the specific points made by each reviewer. In particular, Reviewer 1 makes a strong point that the newly reported genomes either be deposited to GenBank or provided in the supplementary information: the raw reads made available are not sufficient. We will also require a detailed list of your responses to the review comments and a description of the changes you have made in the manuscript.

If you decide to revise the manuscript for further consideration at PLOS Genetics, please aim to resubmit within the next 60 days, unless it will take extra time to address the concerns of the reviewers, in which case we would appreciate an expected resubmission date by email to plosgenetics@plos.org.

We are sorry that we cannot be more positive about your manuscript at this stage. Please do not hesitate to contact us if you have any concerns or questions.

Yours sincerely,

Christopher S. Sullivan

Academic Editor

PLOS Genetics

Kelly Dyer

Section Editor

PLOS Genetics

Reviewer's Responses to Questions

**Comments to the Authors:**

Reviewer #1: Pleolipoviruses represent one of the most widespread and diverse types of viruses infecting hyperhalophilic archaea. In this manuscript, Alarcón-Schumacher and colleagues explore the diversity of pleolipoviruses in metagenomic datasets generated from Australian hypersaline lakes as well as in publicly available databases, namely, IMG/VR. The authors report 164 new pleolipovirus genomes, greatly expanding our appreciation of pleolipovirus diversity and providing new insights into the evolution of this important virus group. The authors also use their data to shed light on certain aspects of virus-host interaction and suggest a revision to pleolipovirus classification. Overall, this is an important study, which following certain modifications outlined below, has the potential to be of great value to the community.

For the results to be actually useful to others, the authors have to either deposit the identified genomes to GenBank (as third party annotations in the case IMG/VR contigs and as new entries for those assembled from the new data) or, at the very least, provide all the genomes as GenBank and fasta formatted files in the supplementary information. Currently, the authors only deposited the raw reads, which is good, but not sufficient. If I or someone else were to reassemble the reads, we would obtain a different set of contigs and would have no way of comparing with those reported in this manuscript. Also note that even if the contigs can be downloaded from IMG/VR, they are not annotated and gene calling could produce different results in other studies. Thus, please provide all the data in one place so that others could build upon it rather than regenerating it from scratch.

The manuscript is too long and could be shortened by removing some generic speculations or moving some sections to the supplement (intended for the biggest pleolipovirus enthusiasts). For instance, “Frequent recombination and gene loss…” is one of such sections, as it presents little new insights and is repetitive with what was said before in the manuscript.

It is not clear why the authors chose not to trim the alignments (line 168), contrary to the widely accepted standards in phylogenetics. This decision might have affected the results and is particularly pertinent to the claims by the authors that VP4 is a bad phylogenetic marker. The impact of trimming is expected to have different impact for short and long proteins, hence, the lack of congruence in phylogenies produced for different proteins. Phylogenetic analyses have to be repeated with properly trimmed alignments to exclude this possibility.

L206-230: The presented data does not quite convince me that “Novel pleolipovirus-like elements from Australian salt lakes reveal the preference for a productive life cycle”. Most of the contigs are very short (cutoff of 2 kb), which might not be sufficient to make assessment of integrated vs extracellular state.

224-230: This part is overly speculative and obscure given that impact of pleolipoviruses on host expression has been shown only for one virus, with all other isolates being apparently rather benign.

L279: If I am not mistaken, vContact genus-level clusters are identified when 3 genes are shared. The authors settled for 2 shared genes. If so, the genus-level calibration might be off.

L290-291: I am really not fond of this sentence: “Altogether, this indicates that the diversity within the Pleolipoviridae family could be greater than previously thought and further challenges the current classification.” It is obvious that diversity within any given virus family is greater than currently sampled and classification has to be constantly revised and expanded once new viruses are discovered. Same comment for statement on L346 – “greater than previously thought” – thought by whom? I personally think that that even after this study we are still very far from appreciating the true genomic diversity of these viruses.

L293: “Betasphaerolipovirus” genus does not exist and the family no longer includes Thermus phages (PMID: 37093734).

L321: Throughout the manuscript the authors keep referring to core genes, referring to HFPV1. First, show a genome map in one of the first figures (upon first mention of core genes). Second, in all other papers on pleolipoviruses HRPV-1 is used as a reference. Given that the latter virus has been studied much more extensively than HFPV1, it does not appear justified to readopt the gene nomenclature – this just brings confusion.

L328 (again on L401): There is no such thing as “very polyphyletic nature” – it is either polyphyletic or not.

L357: I am not convinced that this genome is complete. ITR would strongly suggest the presence of a protein-primed DNA polymerase, like in His2, and there does not seem to be one in this virus. Could the inverted repeats actually represent inverted repeated genes?

L380-383: The choice of examples is rather strange. Why His2, which encodes its own polymerase?

L384: I would suggest refraining from linking HNH to terminase. HNH nucleases perform so many different roles, which are more likely than the one offered.

The Abstract is overly long and tedious. Reads more like an Introduction. Please consider shortening.

All virus taxa should be written in Italics (e.g., line 32).

L40: “one new genera” > “… genus”.

L41: The authors do not demonstrate that genome size is dependent on the ability to integrate. No causality is demonstrated. There might be correlation (should be statistically significant), but not more than that.

L47: “analysis of virion spike proteins reveals that the evolution of the spike protein is driven by the interaction with the cellular surface of the hosts” – this sounds trivial and no data is presented to make a more concrete statement.

L88: “the only isolated member of the Gammapleolipovirus genus” – why was Hardyhisp2 excluded (PMID: 33986080)? This virus has been isolated, not merely sequenced.

Please mention that numerous proviruses related to alpa and betapleolipoviruses have been described previously (e.g PMID: 29361162 etc).

Reviewer #2: Archaea viruses are highly diverse, and with the recent changes to ICTV phage taxonomy classification guidelines, much work needs to be done to reorganize viral taxonomy. The authors identify 129 high-confidence genomes through metagenomic sequencing belonging to the Pleolipoviridae family. They then use this data to propose two new viral genera within the Pleolipoviridae family and predict the origins of these chronic infecting phages.

The paper is very well written with a straightforward story. The authors provide multiple pieces of evidence showing that the Pleolipoviridae phage family needs to be further broken down into additional phage genera using phylogenetic trees, whole genome comparisons, and vConTACT2. It also dives into the evolutionary history of pleolipoviruses and the impact of recombination on the diversity of the viral family. The figures, especially the phylogenetic trees, are very blurry, and parts of the tree are impossible to see. Much work needs to be done on improving figure quality though it could be an issue when uploading the document.

Minor comments

Line 32 Pleolipoviridae in the abstract needs italicized.

Line 34 The sentence has two very different concepts: infection mechanisms and how they evolve. Delete “and” and start a second sentence beginning at interestingly.

Line 85 italicize Betapleolipovirus since that is a genus name.

Line 191, there is a parenthesis “(ORFs… that does not connect to anything. I would recommend deleting that.

Lines 191-200 A lot of what is mentioned is already mentioned in the methods section and could be further summarized.

Line 205. Mentioned 129 genomes were selected for further genome comparisons. There should be a table with all of the genomes with accession numbers referenced.

Line 206, the title is slightly ambiguous. How do the following paragraphs reveal the preference for a productive life cycle? Do you mean a productive, persistent life cycle?

Line 440. The authors mention that the viruses have a narrow host range, but it would be useful to define what they consider is a novel host range, i.e., strain, species, genera, and family-level host ranges. You mention in the following sentence that HFPV-1 infects multiple families, so it is confusing how narrow of a host range the majority of other phages are.

Supplementary Figure 1: It would be helpful to point out how the clusters respond to the proposed phage families.

Major comments

In the abstract, you say you identified 164 new genomes, but in the text, your search yielded 184 non-redundant pleolipovirus genomes (line 246), and 129 of those were used for phylogenetic analysis (line 204). How many novel viruses did you discover, and where does 164 come from?

Figure 2 is very blurry. I am unsure if it is just due to how the picture was uploaded, but I cannot read the conserved features keys, and the entire tree is blurry. The conserved feature key should have the color and shape of the key. It would also be helpful if the internal membrane protein I was not square. When first glancing at Figure A, it looks like the outside ring is showing that the phages belong to alphaplelipovirus.

Figure 3 is also blurry, and it is not easy to read the legend.

Figure 4 This figure is blurry. It would also be helpful to use darker colors if you are going to color-code the tree’s branches. It is not easy to see the yellow and green tree branches.

Figure 6 is blurry.

To establish a new viral taxon, you must have complete viral sequences in the proposed viral genera. It would be useful to add in the text or figures showing the phages with complete genomes. You can refer to this paper for the most recent guidelines for uncultivated virus genome taxonomy classification https://www.nature.com/articles/s41587-023-01844-2 (Adriaenssens et al., Nature Biotechnology, 2023)

It may be worth proposing a subfamily for the beta, gamma, and epsilonviridae phages since the majority do share the rep protein and have circular double-stranded DNA genomes with single-stranded discontinuities.

Based on vConTACT2 and ANI, there appear to be many more genus clusters within the Pleolipovirus-like genomes. It may be worth mentioning that although you may not have enough data to support additional genera, some viruses remain unclassified.

**Have all data underlying the figures and results presented in the manuscript been provided?**

Reviewer #1: **No: **The assembled genome sequences have not been deposited to public databases (only short reads).

Reviewer #2: Yes

PLOS authors have the option to publish the peer review history of their article (what does this mean?). If published, this will include your full peer review and any attached files.

Reviewer #1: No

Reviewer #2: **Yes: **Marissa R Gittrich

---

## [Decision Letter · Decision Letter 1]

26 Sep 2023

Dear Dr Erdmann,

We are pleased to inform you that your manuscript entitled "Revisiting evolutionary trajectories and the organization of the Pleolipoviridae family" has been editorially accepted for publication in PLOS Genetics. Congratulations!

Yours sincerely,

Christopher S. Sullivan

Academic Editor

PLOS Genetics

Kelly Dyer

Section Editor

PLOS Genetics

Comments from the reviewers (if applicable):

Reviewer's Responses to Questions

**Comments to the Authors:**

Reviewer #1: I thank the authors for carefully addressing all of my comments and congratulate them on a well-performed study which will be of great value to the community. I also encourage the authors to submit an official taxonomic proposal to the ICTV in order to officially revise the classification of pleolipoviruses.

**Have all data underlying the figures and results presented in the manuscript been provided?**

Reviewer #1: Yes

PLOS authors have the option to publish the peer review history of their article (what does this mean?). If published, this will include your full peer review and any attached files.

Reviewer #1: **Yes: **Mart Krupovic

**Data Deposition**

http://datadryad.org/submit?journalID=pgenetics&manu=PGENETICS-D-23-00569R1

**Press Queries**

---

## [Editor Report · Acceptance letter]

10 Oct 2023

PGENETICS-D-23-00569R1 

Revisiting evolutionary trajectories and the organization of the Pleolipoviridae family 

Dear Dr Erdmann, 

We are pleased to inform you that your manuscript entitled "Revisiting evolutionary trajectories and the organization of the Pleolipoviridae family" has been formally accepted for publication in PLOS Genetics! Your manuscript is now with our production department and you will be notified of the publication date in due course.

With kind regards,

Anita Estes

PLOS Genetics

On behalf of:
